# Grafting of Amine End-Functionalized Side-Chain Polybenzimidazole Acid–Base Membrane with Enhanced Phosphoric Acid Retention Ability for High-Temperature Proton Exchange Membrane Fuel Cells

**DOI:** 10.3390/molecules29020340

**Published:** 2024-01-10

**Authors:** Guoliang Liu, Hongfei Pan, Shengqiu Zhao, Yadong Wang, Haolin Tang, Haining Zhang

**Affiliations:** 1State Key Laboratory of Advanced Technology for Materials Synthesis and Processing, Wuhan University of Technology, Nr. 122 Luoshi Rd., Wuhan 430070, China; liuguoliang@whut.edu.cn (G.L.); zhaoshengqiu@whut.edu.cn (S.Z.); ywang@whut.edu.cn (Y.W.); thln@whut.edu.cn (H.T.); 2National Energy Key Laboratory for New Hydrogen-Ammonia Energy Technologies, Foshan Xianhu Laboratory, No. 1 Yangming Road, Danzao Town, Nanhai District, Foshan 528200, China; 3Hubei Key Laboratory of Fuel Cell Technology, Wuhan University of Technology, Wuhan 430070, China

**Keywords:** amine functionalized polybenzimidazole, acid uptake, high-temperature proton exchange membrane

## Abstract

A high phosphoric acid uptake and retention capacity are crucial for the high performance and stable operation of phosphoric acid/polybenzimidazole (PA/PBI)-based high-temperature proton exchange membranes. In this work, amine end-functionalized side-chain grafted PBI (AGPBI) with different grafting degrees are synthesized to enhance both the phosphoric acid uptake and the acid retention ability of the accordingly formed membranes. The optimized acid–base membrane exhibits a PA uptake of 374.4% and an anhydrous proton conductivity of 0.067 S cm^−1^ at 160 °C, with the remaining proton conductivity percentages of 91.0% after a 100 h stability test. The accordingly fabricated membrane electrode assembly deliver peak power densities of 0.407 and 0.638 W cm^−2^ under backpressure of 0 and 200 kPa, which are significantly higher than 0.305 and 0.477 W cm^−2^ for the phosphoric acid-doped unmodified PBI membrane under the same conditions.

## 1. Introduction

Proton exchange membrane fuel cells (PEMFCs) have been widely investigated as one of the alternatives to fossil energy in the past decades due to their eco-friendly nature, high energy conversion efficiency, low thermal radiation, and high power output [1,2,3]. Particularly, high-temperature PEMFCs (HT−PEMFCs), which operate above 100 °C, exhibit advantages including improved carbon monoxide tolerance, increased electrode reaction kinetics, and simplified hydrothermal management [4,5,6]. However, the perfluorinated sulfonic acid membrane represented by Nafion cannot meet the requirements of an HT−PEMFC due to its dramatically reduced conductivity caused by severe water loss above 100 °C [7,8,9]. In addition, the European Commission has identified perfluorinated polymers for potential future banning. Therefore, it is necessary to develop novel low-cost hydrocarbon (HC) polymers as viable and safe alternative PEMs that can work at high temperatures under an anhydrous condition for HT−PEMFCs.

The phosphoric acid doped polybenzimidazole (PA/PBI) membrane has been considered as one of the promising HT−PEMs and has made great progress in recent years, in which PA molecules act as efficient proton conductors under anhydrous conditions at a high temperature and PBI chains provide mechanical strength with high chemical and thermal stability [10,11,12,13,14,15,16,17,18]. As a result, the conductivity of a PA/PBI membrane is increased with the acid doping level (ADL) [19,20,21]. The PA/PBI membrane is generally prepared through immersing a PBI membrane obtained by the solution casting method into PA for a certain time to allow PA molecules to enter the membrane. In order to obtain sufficient proton conductivity, great efforts have been made to increase the PA uptake of the PBI membrane [22,23,24,25,26,27,28,29,30].

The commonly used strategy is to prepare side-chain PBI or introduce large steric hindrance groups onto the main chain to attenuate the intermolecular π−π stacking and to improve the fraction free volume of the PBI membrane [31,32,33,34,35,36]. For example, Jana et al. synthesized PBI-based membranes by grafting alkyl side chains with different lengths onto PBI through a N-alkylation reaction. The ADL of the prepared membranes was significantly improved from 6 to 20 due to the damaged hydrogen bonding of PBI by the N-alkylation reaction and the steric hindrance caused by grafting of side chains [31]. They subsequently synthesized a novel type of PBI with nitrogen-rich heterocyclic moieties containing large steric hindrance group. The modification of PBI molecules leads to good solubility and the accordingly formed membrane exhibits an improved ADL from 13 to 21 due to the larger fraction free volume [34]. In addition, in order to reduce the loss of PA molecules and to obtain long-term stable HT−PEM, the introduction of basic or cationic groups into PBI has often been applied to immobilize PA [37,38,39,40,41]. For instance, Tang et al. prepared a series of benzimidazole side chain-grafted PBI. The PBI with basic side chains had both a high ADL of 19 and an excellent PA retention capacity (46% less than that of the original OPBI membrane) owing to the increased free volume of membranes and the enhanced binding energy with PA [37]. Wang et al. prepared side-chain PBI with a quaternary ammonium group through a Menshutkin reaction and they found that the strong interaction between the quaternary ammonium group and PA can significantly improve its PA retention capacity, resulting in a promising remaining conductivity of 0.078 S cm^−1^ at 150 °C for 12 h. [38]. Our previous work also introduced an imidazolium cation into the side chain of PBI and achieved an improvement in both PA uptake and the PA retention capacity [41].

In this work, amine-grafted PBI (AGPBI) was synthesized by the introduction of amino end-functionalized side chains on PBI through an N-alkylation reaction to increase both the fraction free volume and the number of basic groups (Figure 1), aimed at the enhancement of PA adsorption and the PA retention ability. It is thus expected that the accordingly fabricated acid–base membranes exhibit improved anhydrous proton conductivity and long-term stability compared to a PA-doped unmodified PBI membrane. The relationship between the grafting degrees of side chains, proton conductivity, mechanical properties, and stability of AGPBI membranes was systematically studied and the optimum membrane was finally assembled into HT-PEMFC to evaluate its practical application potential.

## 2. Results and Discussion

The chemical structure of original PBI and AGPBI were characterized using ^1^H NMR, as shown in Figure 2a. The resonance peaks of original PBI were located at 7.31–8.31 ppm (7H, 5–9) for the benzene ring and 13.0 ppm (1H, 10) for -NH- group in the imidazole ring, respectively [40]. The response peak at 2.50 ppm is attributed to DMSO solvent. For AGPBI, the four newly appeared resonance peaks at 4.08 (2, 2H), 3.16 (3, 2H), 4.45 (1, 2H) and 7.94 (4, 2H) ppm, attributed to the chemical shift of CPAHCL segments, and the disappearance of the resonance peak of -NH- at 13.0 ppm, indicate the successful synthesis of AGPBI. In addition, the intensity of resonance peaks attributed to the side chain increased with the increase in the grafting degrees (GDs) of AGPBI, and the GD can be calculated by the peak area ratio of the side chain and main chain of AGPBI according to the following equation:(1)$$Grafting\;Degree\;=\;\frac{2\times A_{1}}{A_{8}}\times 100\%;$$

The GD values derived from ^1^H NMR for AG-20, AG-40, AG-70, and AG-100 were 19.7%, 38.1%, 65.4%, and 91.2%, respectively.

FTIR spectroscopy was also carried out to explore the chemical composition of PBI and AGPBI (Figure 2b). All the samples exhibited absorption bands at 1443 and 1601 cm ^−1^, assigned to the stretching vibration of the C=N group and the in-plane deformation vibration of the benzimidazole ring, respectively. After grafting by CPAHCL, the absorption bands attributed to the side-chain methylene appeared at 2852 and 2922 cm^−1^, further confirming the successful synthesis of AGPBI.

The fractional free volumes of the PBI and AG−70 membranes were obtained using positron annihilation lifetime spectra and are listed in Table 1. It can be found that both the free volume radius and density increased with the grafting of the side chain, resulting in the fraction−free volume (FFV) of the AG−70 membrane being significantly higher than that of the PBI membrane, indicating that there was more free volume for phosphoric acid doping.

The PA contact angle measurements were carried out to investigate the affinity of the PBI and AGPBI membrane with PA as show in Figure 3. It is apparent that the contact angle of the AGPBI membranes was lower than that of the original PBI membrane, suggesting that the introduced basic side chain enhances their affinity with PA. Moreover, the contact angle continuously decreased with the increase in GD.

The proton conductivity of PA−doped HT−PEM is closely related to its PA uptake. It can be seen from Table 2 that the PA uptake of the AGPBI membranes was higher than that of the original PBI membrane and it increased with the increase in GD, indicating that the introduction of a side chain is an effective strategy to improve PA uptake, possibly due to the increased fraction free volume of the AGPBI membrane. As a result, the PA uptake of the prepared membranes increased from 202.1% in original PBI to 433.0 in AG-100. Accordingly, the ADL also increased from 8.3 to 22.3 (Table 2). However, the swelling of the prepared membranes also increased with the absorption of PA (Table 2).

Mechanical properties have a significant impact on the practical application of HT−PEMs, since membranes should bear the mechanical load during the fabrication of a membrane electrode assembly and single cells. The stress–strain curves of the prepared PBI and AGPBI membranes before and after PA doping are shown in Figure 4. The derived strength, elongation at break, and fracture energy values are also listed in Table 3. For the undoped membranes, the strength decreased and the elongation at break increased continuously with the increase in GD due to the destroyed hydrogen bonds and the appeared entanglement between the PBI molecules induced by the introduction of side chains. The significant increase in the elongation at break made the fracture energy (the energy required to break the polymer-per-unit volume under tensile load) of the AGPBI membranes more than twice that of the original PBI membrane, with an observed highest value of 7.8 × 10^3^ kJ m^−3^ for the AG−20 membrane. After PA doping, the strength of all membranes decreased and the elongation at break increased remarkably compared with the undoped membranes. The variation in strength and the elongation at break with the GD followed the same trend as the undoped membranes. In addition to the effect caused by the enhanced GD, the improved PA uptake not only further destroyed the hydrogen bonds, but also increased the distance between the polymer molecules, making it more prone to tensile fracture. As a result, the fracture energy of AGPBI was slightly smaller than that of the original PBI membrane.

Proton conductivity is one of the key parameters of HT-PEMs that directly affects the performance of HT-PEMFC. The proton conductivity levels of the PA−doped PBI and AGPBI membranes were measured at a temperature range of 110 to 170 °C under anhydrous conditions, and the results are shown in Figure 5. It is apparent that the proton conductivity of all the tested membranes increased with the increase in both temperature and the GD of side chains. This can be understood as the increase in operating temperature being able to accelerate the movement of protons and PA molecules, while the increase in GD leads to an improvement in PA uptake. For example, the proton conductivity of the AG-70 membrane increased from 0.051 to 0.067 S cm^−1^ with an increasing temperature from 110 to 170 °C. The anhydrous proton conductivity also increased from 0.015 S cm^−1^ for the PA−doped PBI to 0.082 S cm^−1^ for the PA−doped AG−100 membrane at 160 °C, caused by the enhanced PA uptake.

In addition to the proton conductivity, long-term conductivity stability is also critical for HT-PEMs. To elucidate the retention capacity of the amine end-functionalized side chain to PA, the conductivity stability of the prepared membranes was continuously monitored for 100 h, as shown in Figure 6a,b. It is evident that the conductivity loss of all the PA−doped AGPBI membranes was smaller than that of the PA−doped PBI membrane, indicating the less significant PA loss of PA−doped AGPBI compared to the PA−doped PBI membrane. This can be attributed to the introduction of the additional basic amino groups that can electrostatically interact with PA molecules. Furthermore, the conductivity loss of the PA−doped AGPBI membranes decreased first and then increased with the increase in the GD, and the AG−40 membrane showed the lowest conductivity loss ratio. For the alkaline group containing HT-PEMs, PA exists in two ways: bound PA with alkaline groups and free PA stored in molecular pores, which is more prone to loss compared to the former [41]. As the GD increased to 70, the increased grafting density of side-chains could lead to an increased free volume, which in turn resulted in reduced interactions between AGPBI molecules. Thus, the AG−70 membrane exhibited the higher PA uptake and accordingly more PA loss than the sample of AG−40. After 100 h treatment at 160 °C, the remaining conductivity of the prepared membrane was 0.013 S cm^−1^ for PBI, 0.031 S cm^−1^ for AG−20, 0.046 S cm^−1^ for AG−40, 0.060 S cm^−1^ for AG−70, and 0.074 S cm^−1^ for AG−100, corresponding to the remaining conductivity percentages of 88.1%, 90.1%, 92.35%, 91.0%, and 90.3%, respectively.

The PA leaching test, conducted by immersing the membrane in water, was also performed to evaluate the PA retention ability of AGPBI under humidity conditions, as shown in Figure 6c. It can be observed that the PA loss rate of all the membranes in water was much higher than that at 160 °C, while the AG−40 membrane still exhibited the lowest PA weight loss. The AGPBI membranes all showed a better PA retention ability than the original PBI membrane except for the AG−100 membrane, possibly due to its excessive free PA uptake. After 10 h treatment in water, the PA weight remaining of the prepared membranes was 22.9% for PBI, 29.7% for AG−20, 36.4% for AG−40, 31.8% for AG−70 and 18.7% for AG-100, respectively.

For HT-PEMs, its practical application potential should be a comprehensive reflection of mechanical properties, conductivity, and PA retention capacity. In view of the relatively low mechanical properties and PA retention capacity of the PA−doped AGPBI-100 membrane, the AG−70 membrane, with a better comprehensive performance, was selected for further MEA assembly and the original PBI membrane was used for comparison. The single HT-PEMFC performance of the PA−doped PBI and AG−70 membranes was evaluated at 160 °C. The resulting polarization curves, power density curves, and high-frequency resistance response are shown in Figure 7. It can be observed that the power density of the tested membranes improved with the increase in backpressure, owing to the increase in partial pressure of the reaction fuel gas. Moreover, the power density of the fuel cell equipped with the PA-doped AG−70 membrane was much higher than that of the MEA assembled from the PA-doped PBI membrane due to the reduced ohmic polarization voltage loss by the relatively high conductivity of the PA−doped AG−70 membrane, which can be confirmed by the high-frequency resistance of the two single cells. The peak power densities of HT-PEMFC assembled with the PA−doped AG−70 membrane were 0.407 and 0.638 W cm^−2^ under a backpressure of 0 and 200 kPa, respectively, whereas the values for the PA-doped PBI membrane were 0.305 and 0.477 W cm^−2^, respectively. The significantly improved power output demonstrates that the amine end-functionalized side chain-grafted PBI prepared in this work is an effective material to improve the performance of PBI based HT-PEMs. In addition, its performance comparison with the recently reported HT-PEMFCs is also listed in Table 4. It can be noticed that the cell performance in this work is among the highest-performing HT-PEMFCs compared to the recently reported results.

Compared to conductivity stability, the durability of a single fuel cell can better reflect the practical application stability of HT−PEMs. Durability tests of the PBI and AG−70 membranes at 160 °C at 0.5 A cm^−2^ under back pressures of 200 kPa were performed, the results of which are shown in Figure 8. After continuous operation for 50 h, the output voltage of a single HT−PEMFC assembled with an AG-70 membrane was slightly reduced, with a relatively low decay rate of 38.4 μV h^−1^, which is significantly lower than the value of 84.3 μV h^−1^ of the PBI-assembled single cell, indicating its practical application potential in highly stable HT−PEMs.

## 3. Experimental

### 3.1. Materials and Chemicals

Poly(4,4′-diphenylether-5,5′-bibenzimidazole) (PBI) was obtained from Tianzhihong Plastic Co., Ltd. (Dongguan, China), with a M_n_ of 25 kDa. Sodium hydride (NaH) and 3-chloropropylamine hydrochloride (CPAHCL) were purchased from Aladdin Inc. (Shanghai, China). *N*,*N*-dimethylacetamide (DMAc, A.R.), phosphoric acid (PA, A.R.), acetone (A.R.), isopropanol (A.R.), and ethanol (A.R.) were received from Sinopharm Chemical Reagent Corp. (Shanghai, China).

### 3.2. Synthesis of Amine-Grafted Polybenzimidazole

A total of 5 g (25 mmol of -NH- groups) of PBI was dissolved in 200 mL DMAc at 80 °C under nitrogen atmosphere protection. After 2 g of NaH (excessive) was added into the PBI solution for pre-activating with the aid of avoiding the self-condensation of CPAHCL, the mixture was magnetically stirred for 12 h at 45 °C. Subsequently, a desired amount of CPAHCL (5, 10, 17.5, and 25 mmol) was added into the mixture and stirred at 90 °C for 12 h. After cooling down to room temperature, the solution was poured into ice acetone to precipitate amine-grafted polybenzimidazole (AGPBI). The resulting AGPBI membranes were labeled as AG−20, AG−40, AG−70, and AG−100 according to their theoretical grafting degrees. The yield of amine-grafted polybenzimidazole was 92.1% for AG−20, 89.3% for AG−40, 84.7% for AG−70 and 78.2% for AG−100, respectively. The AGPBI products were washed with ethanol and water three times to completely remove the residual reactant, and finally dried at 80 °C for 24 h.

### 3.3. Preparation of Phosphoric Acid Doped AGPBI Membranes

A weight of 0.25 g of AGPBI was dissolved in 20 mL DMAc at 80 °C to form a homogeneous solution. The solution was poured onto a glass plate and dried at 80 °C for 24 h to prepare the AGPBI membrane. The AGPBI membrane was then taken from the plate and treated at 150 °C to remove the residual solvent. For the PA doping process, AGPBI membranes were immersing in 85 wt.% PA at 60 °C for 1 h. Subsequently, the membranes were taken out and we wiped the redundant PA away with filter paper. The final membranes were dried at 90 °C for 5 h and stored in a desiccator. For comparison, the PA-doped pristine PBI membrane was prepared using the same process. It should be pointed out that in order to obtain the same thickness of the final PA-doped membranes, the undoped OPBI and AGPBI membranes were prepared by descending thickness as the modification degree increased owing to the increased PA swelling. Subsequently, the membranes with different thicknesses were immersed in PA to achieve a consistent thickness of 60 ± 6 μm for further testing.

### 3.4. Characterization

A nuclear magnetic spectrometer (Bruker AVANCE III HD 500MHz, Billerica, MA, USA) was applied to explore the chemical shifts of the original PBI and AGPBI using DMSO-d_6_ as the solvent and tetramethylsilane as the internal standard for calibration. Fourier-transform infrared (FTIR) spectra were recorded on a Thermo Scientific Nicolet 6700 spectrophotometer (Waltham, MA, USA) to investigate the chemical composition of the original PBI and AGPBI in a wavenumber range from 4000 to 400 cm^−1^, with a resolution of 4 cm^−1^. The Labthink Model XLW-EC-A tensile tester (Medford, MA, USA) was applied to measure the stress–strain curves of the original PBI and AGPBI membranes using a tensile rate of 50 mm min^−1^. PA contact angle measurements of the original PBI and AGPBI membranes were conducted on a POWEREACH JC2000C tester (Shanghai, China).

### 3.5. PA Uptake, Swelling and Acid Doping Level

The PA uptake represents the percentage increase in mass and the swelling corresponds to volume growth rate after the doping process. The PA uptake, swelling, and acid doping level can be calculated as follows:(2)$$PA\;uptake\;(\%)\;=\;\frac{m_{a}-{\;m}_{b}}{m_{b}}\times 100\%;$$
(3)$$Swelling\;(\%)\;=\;\frac{V_{a}-{\;V}_{b}}{V_{b}}\times 100\%;$$
(4)$$ADL\;=\;\frac{{(m}_{a}-{{\;m}_{b})/M}_{PA}}{m_{b}/M_{AGPBI}};$$
where the $m_{b}$ and $m_{a}$ are the mass of the prepared membranes before and after PA doping, and $V_{b}$ and $V_{a}$ are the corresponding volume, respectively. $M_{PA}$ and $M_{AGPBI}$ are the molecule weight of PA and AGPBI, respectively. The membrane samples were cut into 5 × 5 cm size for the PA uptake, swelling and ADL test. For the swelling test, the length and width were firstly tested for three and five times at different positions, respectively. Subsequently, the PA uptake, swelling and ADL of all samples were tested three times. The average value was taken and the error was recorded.

### 3.6. Proton Conductivity and PA Retention Capacity

The proton conductivity (*σ*) was performed on a Princeton versaSTAT3 electrochemical workstation in a frequency range from 1 MHz to 1 Hz, with an amplitude of 5 mV at a temperature range of 110 to 170 °C without humidification. The samples were cut into 3 × 2 cm and each membrane was tested three times. The values were obtained from the following:(5)$$\sigma =\frac{L}{R\;\times \;A}$$
where L, R and A are the effective length, resistance, and cross-sectional area of the tested membranes, respectively.

The long-term PA retention capacity of the PA-doped membranes were characterized by two methods. The former reflected the PA retention capacity by monitoring the conductivity changes at the operating temperature (160 °C). The latter involved placing the membrane in deionized water, then drying and recording the remaining mass every hour to calculate the loss rate of PA in the membrane.

### 3.7. Positron Annihilation Lifetime Measurements

The positron annihilation experiments were performed as reported in the literature [40]. The long-life component *τ*_3_ was due to o-Ps pick-up annihilation in free volume holes in the membrane. The radius (*R*), average volume (*V*) and fraction-free volume (FFV) can be calculated from the following equations, respectively [41]:(6)$$\tau_{3}=0.5{\left[1-\frac{R}{R+\Delta R}+\frac{1}{2\pi }\sin\left(\frac{2\pi R}{R+\Delta R}\right)\right]}^{-1}$$
(7)$$V=\frac{4}{3}{\pi R}^{3}$$
(8)$$ƒ=AVI_{3}$$
where the Δ*R* = 0.1656 nm is the thickness of the o-Ps wave function overlapping with the homogeneous electron layer; *A* can be approximately regarded as 0.0018 Å^−3^.

### 3.8. Membrane Electrode Assembly and Fuel Cell Performance

The membrane electrode assembly (MEA) was fabricated according to our previous study, with an effective area of 25 cm^2^ and Pt loading of 1.0 mg cm^−2^ for both electrodes [41]. The gas flow rates for the anode and cathode were 1000 mL min^−1^ of hydrogen and 2500 mL min^−1^ of air, respectively. The Hephas Energy HTS-125s fuel cell test station was used to monitor the polarization curve and high-frequency resistance (HFR) under a backpressure of 0 and 200 kPa.

## 4. Conclusions

The amine end-functionalized side chain-grafted PBI (AGPBI) was synthesized using an N-alkylation reaction with 3-chloropropylamine hydrochloride and high temperature proton exchange membranes were fabricated using the solvent casting of AGPBI, followed by a phosphoric acid doping process. Due to the improved affinity with phosphoric acid induced by the grafted amino groups, the prepared acid–base AGPBI membranes showed a higher conductivity and enhanced phosphoric acid retention capacity compared to the unmodified PBI membrane. As the optimal HT-PEM, the AG-70 membrane has a proton conductivity of 0.067 S cm^−1^ and 91.0% phosphoric acid retention after a 100 h stability test. The peak power density values of a single cell assembled from a phosphoric acid-doped AG-70 membrane are 0.407 and 0.638 W cm^−2^ under a backpressure of 0 and 200 kPa. In addition, the decrease in the mechanical strength of AGPBI was observed compared with the unmodified PBI membrane. Although the fracture energy of AG-70 membrane slightly increased, it is still necessary to further improve its mechanical strength and our further work is in process to crosslink the AGPBI.

## Figures and Tables

**Figure 1 molecules-29-00340-f001:**
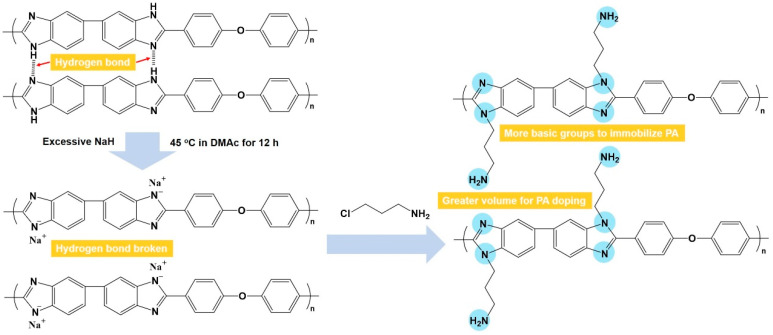
Synthetic scheme of amine-grafted polybenzimidazole.

**Figure 2 molecules-29-00340-f002:**
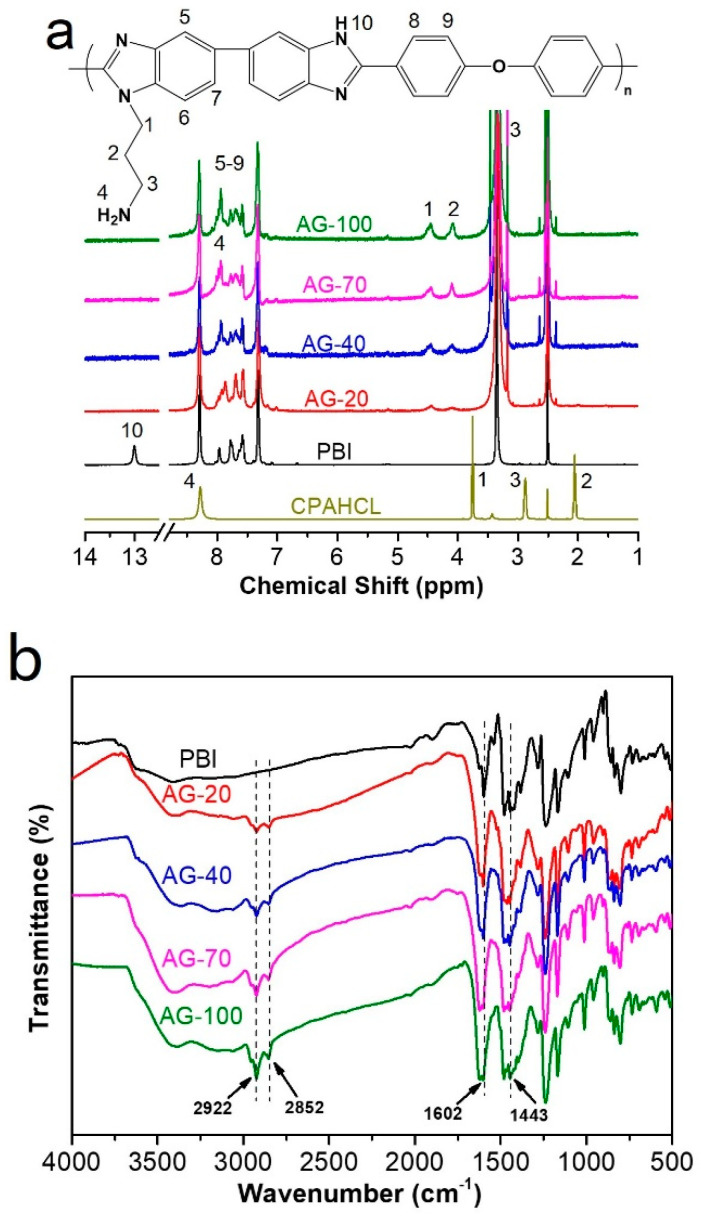
H NMR (**a**) and FTIR (**b**) spectra of PBI and AGPBI.

**Figure 3 molecules-29-00340-f003:**
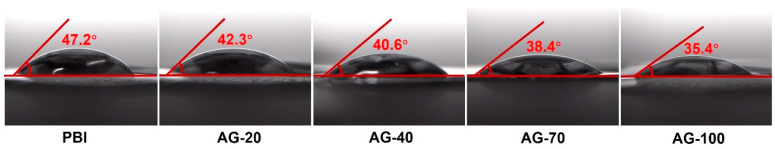
PA contact angle of the PBI and AGPBI membranes.

**Figure 4 molecules-29-00340-f004:**
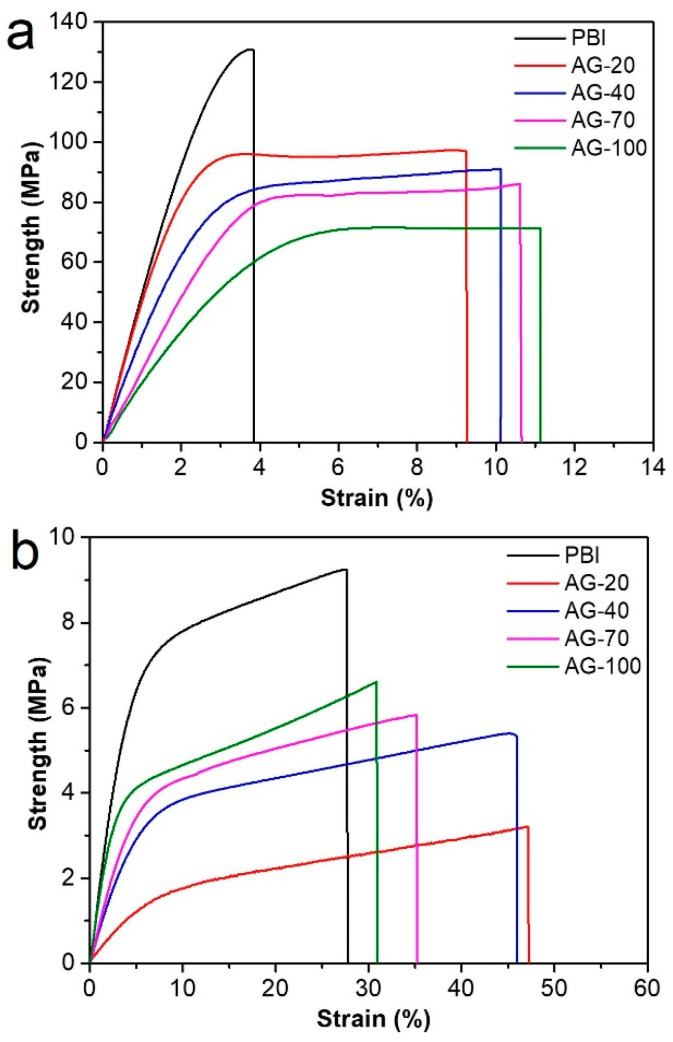
Stress–strain curves of undoped (**a**) and PA-doped (**b**) membranes.

**Figure 5 molecules-29-00340-f005:**
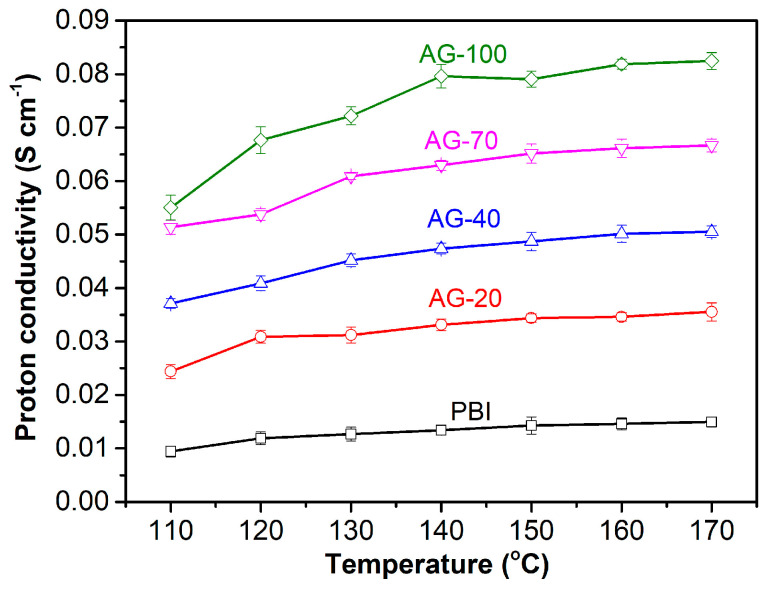
Anhydrous proton conductivity of PA−doped PBI and AGPBI membranes at temperature range from 110 to 170 °C.

**Figure 6 molecules-29-00340-f006:**
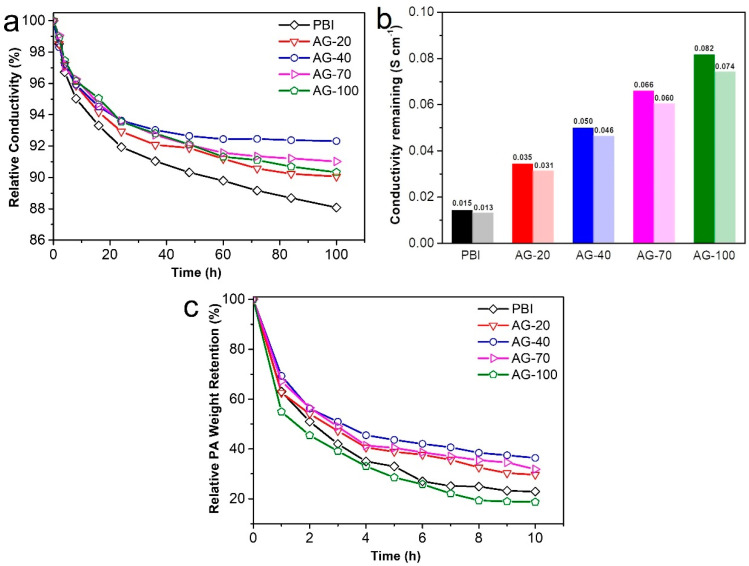
Relative conductivity of the prepared membranes at 160 °C for 100 h (**a**), conductivity remaining after 100 h test (**b**) and PA weight loss in water (**c**).

**Figure 7 molecules-29-00340-f007:**
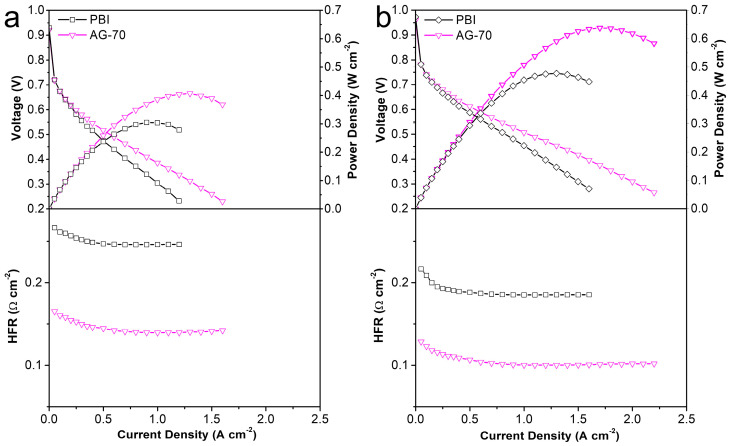
Polarization, power density, and high-frequency resistance curves of MEAs assembled from PA-doped PBI and AG-70 membranes at 160 °C under a backpressure of 0 (**a**) and 200 kPa (**b**).

**Figure 8 molecules-29-00340-f008:**
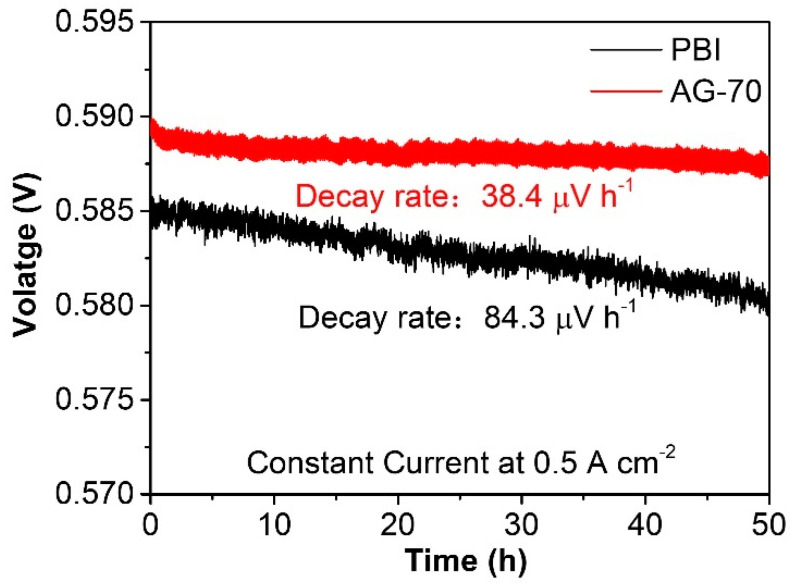
Durability of AG−70 membrane at 160 °C at 0.5 A cm^−2^ under back pressures of 200 kPa.

**Table 1 molecules-29-00340-t001:** o-Ps lifetimes, intensities, radii and fractional free volumes of the PBI and AG−70 membranes.

Samples	*τ_3_* (ns)	*I_3_* (%)	*R* (Å)	FFV
PBI	1.799	9.2	4.59	6.72
AG-70	1.901	10.3	4.97	9.53

**Table 2 molecules-29-00340-t002:** GD, PA uptake, ADL, and swelling of the unmodified PBI and AGPBI membranes.

Samples	GD (%)	PA Uptake (wt.%)	ADL	Swelling (%)
PBI	0	202 ± 12.3	8 ± 0.5	131 ± 7.9
AG-20	19.7	288 ± 11.8	12 ± 0.5	166 ± 8.8
AG-40	38.1	330 ± 16.9	16 ± 0.8	191 ± 11.2
AG-70	65.4	374 ± 23.5	18 ± 1.1	235 ± 18.7
AG-100	91.2	433 ± 22.8	22 ± 1.2	284 ± 17.5

**Table 3 molecules-29-00340-t003:** Mechanical properties of undoped and PA-doped membranes.

Samples	Strength (MPa)		Elongation at Break (%)		Fracture Energy (10^3^ kJm^−3^)	
	Undoped	Doped	Undoped	Doped	Undoped	Doped
PBI	130.8 ± 7.1	8.7 ± 1.5	3.8 ± 0.5	20.5 ± 2.7	3.1 ± 0.2	1.7 ± 0.2
AG-20	97.3 ± 7.8	6.6 ± 1.8	9.3 ± 0.6	30.9 ± 2.2	7.8 ± 0.7	1.5 ± 0.1
AG-40	91.0 ± 5.2	5.8 ± 2.3	10.1 ± 0.7	35.2 ± 3.1	7.6 ± 0.6	1.6 ± 0.2
AG-70	86.0 ± 3.8	5.4 ± 1.7	10.6 ± 0.6	46.0 ± 2.9	7.3 ± 0.6	1.9 ± 0.2
AG-100	71.5 ± 8.8	3.2 ± 1.8	11.1 ± 0.9	47.2 ±3.9	6.4 ±1.1	1.0 ± 0.1

**Table 4 molecules-29-00340-t004:** Performance comparison with the recently reported HT-PEMFCs.

Membrane	Fuel Gas	Backpressure (kPa)	Pt Loading (mg cm^−2^) (Cathode)	Peak Power Density (W cm^−2^)	Ref.
AG-70	H_2_, Air	0	1	0.409	**This work**
		200	1	0.638	
CTFs-OPBI	H_2_, O_2_	0	1	0.534	[12]
AmPBI-Car-5	H_2_, Air	0	1.5	0.216	[33]
PBI-Sc-5	H_2_, Air	0	1.5	0.411	[38]
NbPBI-TPAm	H_2_, Air	0	1.5	0.385	[42]
PBI/sGO-2	H_2_, Air	0	1	0.364	[43]
PBI/1Mus	H_2_, Air	0	1	0.586	[44]
L-10	H_2_, Air	0	0.6	0.438	[45]
IPyPBIs	H_2_, O_2_	0	0.6	0.28	[46]
QPANI-OPBI	H_2_, O_2_	0	0.6	0.459	[47]
DPBI-10PVBC	H_2_, O_2_	0	0.4	0.405	[48]
NPBI	H_2_, O_2_	0	0.5	0.632	[49]
1%-PBI	H_2_, O_2_	0	0.6	0.597	[50]
PPA-PBI	H_2_, Air	0	1	~0.6	[51]
PPA-2,5-PPBI	H_2_, O_2_	0	1	~0.6	[52]
PA-PBI	H_2_, Air	300	1	~0.4	[53]

## Data Availability

Data are contained within the article.

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
