# Peer review of "Grafting of Amine End-Functionalized Side-Chain Polybenzimidazole Acid–Base Membrane with Enhanced Phosphoric Acid Retention Ability for High-Temperature Proton Exchange Membrane Fuel Cells"

_molecules, 2024, doi:10.3390/molecules29020340_

Round 1

Reviewer 1 Report

Comments and Suggestions for Authors

Dear Authors

General comments

The PBI membrane is modified with amine groups, so it is more convenient to replace "grafting" with "modified".

The resultant PA-doped AG membranes are "Acid-Base" membranes due to the interaction between Phosphoric acid and amine base, so it is recommended to reflect such description throughout the text and accordingly modify the title.

An essential characterization such as TGA, SEM, pore size, and volume analyses are missed. It should be performed and correlated to the proton conductivity, PA uptake, ADP, and durability results.

To verify the authors' basic concept about the role of the introduced amine groups, it is logical to test different length amine modifying agents such as ethyl amine, butane amine, and propane amine. 

The dropped PA can act as an ionic cross-linker for the introduced amine groups. The authors need to discuss such probability.   

 specific comments

In section 2.2. Synthesis of Amine-Grafted Polybenzimidazole, the authors mentioned that"The yield of amine-grafted polybenzimidazole was 92.1% for AG-20, 89.3% for AG-40, 84.7% for AG-20 and 78.2% for 95 AG-100, respectively". The authors should declare sample codes means; AG-20, AG-40, AG-40. Furthermore, how come the same sample AG-20 has two yields of amine grafting; 92.1% for AG-20, and AG-20 and 78.2%? Please revise and correct.

In section 3. Results and Discussion, the authors mentioned on page 7 a discussion of the long-term conductivity stability. They referred to the obtained behavior as the free volume increase. To support their claim, pores volume measurement of the developed AG membranes should be performed and correlated to the obtained results in Figure 6. SEM analysis of AG membranes cross-section will be supportive. The same for the discussion of PA loss due to water leaching; Figure 6c.

In the evaluation of the durability of the developed phosphoric acid-doped AG-70 membrane (Figure 8), the blank PA-doped PBI membrane should be included for better comparison. 

I can recommend the publication of the manuscript after considering the above-mentioned comments.

Greetings

Comments on the Quality of English Language

A minor revision is needed. 

Author Response

Response to reviewer #1

General comments:

  1. The PBI membrane is modified with amine groups, so it is more convenient to replace "grafting" with "modified".

Response: We thank the reviewer for the comment. Although "modification" is also applicable to describe our work, "grafting" may be a closer term because the amino containing side chains are chemically attached onto PBI mainchains through the N-alkylation reaction.

  1. The resultant PA-doped AG membranes are "Acid-Base" membranes due to the interaction between Phosphoric acid and amine base, so it is recommended to reflect such description throughout the text and accordingly modify the title.

Response: We sincerely thank the reviewer for the valuable suggestion. We have reflected the description of "acid-base" throughout the text and accordingly revised the title to " Grafting of amine end-functionalized side-chain polybenzimidazole acid-base membrane with enhanced phosphoric acid retention ability for high temperature proton exchange membrane fuel cells".

  1. An essential characterization such as TGA, SEM, pore size, and volume analyses are missed. It should be performed and correlated to the proton conductivity, PA uptake, ADP, and durability results.

Response: We apologize for the missed characterization results and they were added in Table 1 in the revised work. The according discussion was also made in the revised manuscript in page 9, page 11 as highlighted with red colors.

  1. To verify the authors' basic concept about the role of the introduced amine groups, it is logical to test different length amine modifying agents such as ethyl amine, butane amine, and propane amine.

Response: We thank the reviewer for the valuable comments. We definitely agree that it is reasonable to investigate amino modified PBI with different side chain lengths. However, the focus of present work is to determine the effects of different degrees of amino side chain modification on acid doping and single cell performance, etc. We believe that the investigation of the impact of different side chain lengths should be an individual work. Nevertheless, the comment by the reviewers provides guidance for our future work direction.

  1. The dropped PA can act as an ionic cross-linker for the introduced amine groups. The authors need to discuss such probability.

Response: We very much appreciate the reviewer for the valuable suggestion.   When the molar ratio of phosphoric acid to amino group is less than 1, this ionic-crosslinking may theoretically occur. However, in this work, the molar ratio of phosphoric acid to amino group is much higher than 1 and this ionic-crosslinking may not occur.

Point by point comments:

  1. In section 2.2. Synthesis of Amine-Grafted Polybenzimidazole, the authors mentioned that"The yield of amine-grafted polybenzimidazole was 92.1% for AG-20, 89.3% for AG-40, 84.7% for AG-20 and 78.2% for 95 AG-100, respectively". The authors should declare sample codes means; AG-20, AG-40, AG-40. Furthermore, how come the same sample AG-20 has two yields of amine grafting; 92.1% for AG-20, and AG-20 and 78.2%? Please revise and correct.

Response: We are sorry for the confused presentation. This has been declared in section 2.3 in the revised manuscript as highlighted with red colors. In order to avoid reading difficulties, we have changed the description to “The resulted AGPBI membranes were labeled as AG-20, AG-40, AG-70, and AG-100 according to their theoretical grafting degrees.” in section 2.2 in page 6 in the revised version as highlighted with red colors. In addition, the “84.7% for AG-20” is an error and we have corrected it to “84.7% for AG-70” in the revised manuscript.

  1. In section 3. Results and Discussion, the authors mentioned on page 7 a discussion of the long-term conductivity stability. They referred to the obtained behavior as the free volume increase. To support their claim, pores volume measurement of the developed AG membranes should be performed and correlated to the obtained results in Figure 6. SEM analysis of AG membranes cross-section will be supportive. The same for the discussion of PA loss due to water leaching; Figure 6c.

Response: We thank the reviewer for the comment. We clarified in the revised manuscript that “For the alkaline group containing HT-PEMs, PA exists in two ways: bound PA with alkaline groups and free PA stored in molecular pores, which is more prone to loss compared to the former [41]. As the GD increases to 70, the increased grafting density of side-chains can lead to the increased free volume, which in turn results in the reduced interactions between AGPBI molecules. Thus, the AG-70 membrane exhibited the higher PA uptake and the accordingly more PA loss than the sample of AG-40” in page 16 in the revised manuscript as highlighted with red colors. We are trying to explain the reason for the slight decrease in conductivity stability starting from the AG-70 membrane. Nevertheless, the free fraction volume has still been measured to explain the high ADL of AGPBI membranes. The relevant discussions and conclusions have also been added to the revised manuscript as highlighted with red colors in page 11.

  1. In the evaluation of the durability of the developed phosphoric acid-doped AG-70 membrane (Figure 8), the blank PA-doped PBI membrane should be included for better comparison.

Response: We appreciate the reviewer for the valuable suggestion. The single cell durability of the blank PA-doped PBI membrane has been carried out for comparison in the revised manuscript. The results were presented as Figure 8 and the according discussion was made in page 20 as highlighted with red colors in the revised manuscript.

Reviewer 2 Report

Comments and Suggestions for Authors

molecules-2775180

Grafting of Amine End-Functionalized Side-Chain on Polybenzimidazole with Enhanced Phosphoric Acid Retention Ability for High Temperature Proton Exchange Membrane Fuel Cells

This study synthesizes amine end-functionalized side chain grafted PBI for application in high-temperature proton exchange membrane fuel cells (PEMFCs). The introduction of grafted amino groups enhances the affinity for phosphoric acid, leading to improved ionic conductivity and enhanced phosphoric acid retention. Upon transitioning from PBI to the modified PBI, the fuel cell's peak power density increases from 0.477 to 0.638 W/cm2 at 160°C and 200 kPa. However, attention is required to enhance the mechanical strength of the modified PBI.

The authors have conducted thorough materials characterizations and electrochemical performance assessments on the modified PBI membranes, crucial for high-temperature PEMFCs. The manuscript is well-prepared, and the reviewer recommends acceptance with the suggestion that the authors address the following comments and suggestions in their revision.

·        Line 95 – Provide an explanation for the numerical designations in AG-20/40/70/100. Correct the second instance of AG-20 to AG-70.

·        Critical information regarding the membrane is absent. Clearly state the thicknesses of both the PBI and modified membranes, and elucidate how well these thicknesses are controlled. If possible, include SEM images offering a cross-sectional view of the membranes.

·        Given the challenges in volume measurements, which may contribute to relatively high uncertainty in the swelling rate (Eq. 2), elaborate on how samples are prepared for volume measurement and specify the measurement accuracy of the volumes.

·        Lines 156-157: Clarify the estimation of grafting degrees (GDs) from the NMR data.

·        In Fig. 6 (a) and (c), adjust the y-axis labels to “relative conductivity” and “relative PA weight retention” instead of xxx loss. Correspondingly, on line 251, modify “the PA weight loss of the …” to “the PA weight remaining of the ….”

·        Fig. 8: While it is commendable to observe a durability test of the AG-70 membrane in HT-PEMFC, consider enhancing the strength of the comparison by including the durability performance of the PBI membrane under the same conditions in the same figure.

Comments on the Quality of English Language

Acceptable.

Author Response

Response to reviewer #2

General comments:

  1. This study synthesizes amine end-functionalized side chain grafted PBI for application in high-temperature proton exchange membrane fuel cells (PEMFCs). The introduction of grafted amino groups enhances the affinity for phosphoric acid, leading to improved ionic conductivity and enhanced phosphoric acid retention. Upon transitioning from PBI to the modified PBI, the fuel cell's peak power density increases from 0.477 to 0.638 W/cm2 at 160°C and 200 kPa. However, attention is required to enhance the mechanical strength of the modified PBI.

 Response: We would like to thank the reviewer for these valuable and positive comments. Although the mechanical properties of the prepared membrane in this work decrease with increase of the degree of modification and phosphoric acid doping level, it can still meet requirements of the application in fuel cells. Nevertheless, according to the reviewer’s suggestion, we will enhance the mechanical properties of these membranes by cross-linking or adding reinforced skeleton in our future work.

  1. The authors have conducted thorough materials characterizations and electrochemical performance assessments on the modified PBI membranes, crucial for high-temperature PEMFCs. The manuscript is well-prepared, and the reviewer recommends acceptance with the suggestion that the authors address the following comments and suggestions in their revision.

Response: We would like to thank the reviewer for these valuable and positive comments.

Point by point comments:

  1. Line 95 – Provide an explanation for the numerical designations in AG-20/40/70/100. Correct the second instance of AG-20 to AG-70.

Response: We are sorry for the missed description. This has been clarified in the revised manuscript as “The resulted AGPBI membranes were labeled as AG-20, AG-40, AG-70, and AG-100 according to their theoretical grafting degrees.” in section 2.2 in page 6 in the revised version as highlighted with red colors

  1. Critical information regarding the membrane is absent. Clearly state the thicknesses of both the PBI and modified membranes, and elucidate how well these thicknesses are controlled. If possible, include SEM images offering a cross-sectional view of the membranes.

Response: We apologize for the unclear description. The thickness of the prepared PA doped membranes is 60 ± 6 μm (the cross-sectional SEM images has listed as below) and the membrane thickness control strategy has also been added to the revised manuscript as highlighted with red color in page 7.

Figure R1       Cross-sectional SEM images of PA-doped PBI and AGPBI membranes.

  1. Given the challenges in volume measurements, which may contribute to relatively high uncertainty in the swelling rate (Eq. 2), elaborate on how samples are prepared for volume measurement and specify the measurement accuracy of the volumes.

Response: We thank the reviewer for the comments. The detailed process for sample preparation and PA swelling test of membranes have been added to the revised manuscript as highlighted with red color in page 7 and page 8.

  1. Lines 156-157: Clarify the estimation of grafting degrees (GDs) from the NMR data.

Response: We very much appreciate the reviewer’s suggestion. The calculation process of GD values obtained from 1H NMR results has been added to the revised manuscript as highlighted with red color in page 10 and page 11.

  1. In Fig. 6 (a) and (c), adjust the y-axis labels to “relative conductivity” and “relative PA weight retention” instead of xxx loss. Correspondingly, on line 251, modify “the PA weight loss of the …” to “the PA weight remaining of the ….”

Response: We thank the reviewer for the suggestion. This has been corrected in the revised manuscript.

  1. Fig. 8: While it is commendable to observe a durability test of the AG-70 membrane in HT-PEMFC, consider enhancing the strength of the comparison by including the durability performance of the PBI membrane under the same conditions in the same figure.

Response: We very much appreciate the suggestion arisen from the reviewer. The single cell durability of the blank PA-doped PBI membrane has been carried out for comparison in the revised manuscript (Figure 8) and the according discussion was made in page 20 as highlighted with red colors in the revised manuscript.

Reviewer 3 Report

Comments and Suggestions for Authors

Creating phosphoric acid-doped polybenzimidazole (PA-PBI) membranes with both high proton conductivity and robust mechanical strength presents a significant challenge, demanding a straightforward and scalable preparation method. In this work, the authors enhanced both phosphoric acid uptake and acid retention ability by grafting amine end-functionalized side chains from PBI (AGPBI) with varying degrees of grafting, all while minimizing any significant compromise to the mechanical properties. We would like to suggest this manuscript be accepted for publish in Molecules after addressing the following minor revision issues.

1.            In the introduction, a. suggests to highlight the urgency in seeking safer alternatives to perfluorinated polymers. The European Commission has identified perfluorinated polymers for potential future banning. This regulatory attention underscores the critical necessity for exploring low-cost hydrocarbon (HC) polymers as viable and safer alternatives. b. For a more scientifically rigorous evaluation of recent published work on phosphoric acid (PA) uptake and retention capacity, providing specific numerical values is preferable over terms like "excellent," "high," or "improve."

2.            The experiment section is lacking detailed information regarding the conductivity measurement.

3.            The short-term stability testing duration is insufficient. We recommend extending the testing period to 200 hours.

4.            On page 5, line 177, the sentence is unrelated to this section.

5.            Please review all the data in Table 1, specifically focusing on Accuracy and Precision. For instance, the swelling (%) values should be rounded to a single digit, such as 130% and 166%. 

Author Response

Response to reviewer #3

General comments:

Creating phosphoric acid-doped polybenzimidazole (PA-PBI) membranes with both high proton conductivity and robust mechanical strength presents a significant challenge, demanding a straightforward and scalable preparation method. In this work, the authors enhanced both phosphoric acid uptake and acid retention ability by grafting amine end-functionalized side chains from PBI (AGPBI) with varying degrees of grafting, all while minimizing any significant compromise to the mechanical properties. We would like to suggest this manuscript be accepted for publish in Molecules after addressing the following minor revision issues.

Response: We would like to thank the reviewer for these valuable and positive comments.

Point by point comments:

  1. In the introduction, a. suggests to highlight the urgency in seeking safer alternatives to perfluorinated polymers. The European Commission has identified perfluorinated polymers for potential future banning. This regulatory attention underscores the critical necessity for exploring low-cost hydrocarbon (HC) polymers as viable and safer alternatives. b. For a more scientifically rigorous evaluation of recent published work on phosphoric acid (PA) uptake and retention capacity, providing specific numerical values is preferable over terms like "excellent," "high," or "improve."

Response: We sincerely thank the reviewer for the valuable suggestion. They were all corrected and modified in the revised manuscript as highlighted with red colors.

  1. The experiment section is lacking detailed information regarding the conductivity measurement.

Response: We apologize for the less-detailed description on the experimental process. The detailed information for the conductivity measurement has been added into the revised manuscript as highlighted with red color in page 8.

  1. The short-term stability testing duration is insufficient. We recommend extending the testing period to 200 hours.

Response: We apologize for not being able to perform longer durability tests owing to the relative shortage of testing conditions and personnel reasons. However, in order to further verify the conclusion of this work, we also conducted short-term durability tests on single cell assembled with unmodified PA/PBI membranes. The relevant data and discussion were added to the revised manuscript as highlighted with red colors in page 20.

  1. On page 5, line 177, the sentence is unrelated to this section.

Response: We apologize for the mistake and the un-related sentence has been deleted.

  1. Please review all the data in Table 1, specifically focusing on Accuracy and Precision. For instance, the swelling (%) values should be rounded to a single digit, such as 130% and 166%. 

Response: We sincerely thank the reviewer for the suggestion. The data in Table 1 has been rounded to a single digit in the revised manuscript as highlighted with red color.

Round 2

Reviewer 1 Report

Comments and Suggestions for Authors

Dear Authors

The revised version has considered the raised comments.

Accordingly, I can recommend the revised manuscript for publication.

Greetings,

Comments on the Quality of English Language

A minor revision is recommended.